# Dopamine Receptor Ligand Selectivity—An In Silico/In Vitro Insight

**DOI:** 10.3390/biomedicines11051468

**Published:** 2023-05-17

**Authors:** Lukas Zell, Alina Bretl, Veronika Temml, Daniela Schuster

**Affiliations:** Department of Pharmaceutical and Medicinal Chemistry, Institute of Pharmacy, Paracelsus Medical University, 5020 Salzburg, Austria; lukas.zell@pmu.ac.at (L.Z.); alina.bretl@pmu.ac.at (A.B.); veronika.temml@pmu.ac.at (V.T.)

**Keywords:** dopamine receptor, subtype selectivity, GPCR, in silico, molecular docking, secondary binding pocket, in vitro, HTRF

## Abstract

Different dopamine receptor (DR) subtypes are involved in pathophysiological conditions such as Parkinson’s Disease (PD), schizophrenia and depression. While many DR-targeting drugs have been approved by the U.S. Food and Drug Administration (FDA), only a very small number are truly selective for one of the DR subtypes. Additionally, most of them show promiscuous activity at related G-protein coupled receptors, thus suffering from diverse side-effect profiles. Multiple studies have shown that combined in silico/in vitro approaches are a valuable contribution to drug discovery processes. They can also be applied to divulge the mechanisms behind ligand selectivity. In this study, novel DR ligands were investigated in vitro to assess binding affinities at different DR subtypes. Thus, nine D_2_R/D_3_R-selective ligands (micro- to nanomolar binding affinities, D_3_R-selective profile) were successfully identified. The most promising ligand exerted nanomolar D_3_R activity (Ki = 2.3 nM) with 263.7-fold D_2_R/D_3_R selectivity. Subsequently, ligand selectivity was rationalized in silico based on ligand interaction with a secondary binding pocket, supporting the selectivity data determined in vitro. The developed workflow and identified ligands could aid in the further understanding of the structural motifs responsible for DR subtype selectivity, thus benefitting drug development in D_2_R/D_3_R-associated pathologies such as PD.

## 1. Introduction

G-protein coupled receptors (GPCRs) are one of the most prominent protein families targeted in drug research. Currently, they are represented by 475 approved (by the FDA) drugs acting on 108 different GPCRs [1]. Sixty-five of them target an essential sub-group of GPCRs, the dopamine receptor (DR) family, consisting of the subtypes 1, 2, 3, 4 and 5 (D_1_R, D_2_R, D_3_R, D_4_R and D_5_R, respectively) [2]. The DR family is divided into D_1_like-(D_1_R and D_5_R) and D_2_like receptors (D_2_R, D_3_R and D_4_R) and plays a crucial role in physiological processes such a motoric function, cognition, sleep and memory [3]. However, it is also involved in many devastating diseases of the central nervous system (CNS) such as Parkinson’s Disease (PD), schizophrenia and bipolar disorders. DR-targeting drugs act in different ways acting as, e g., agonists in PD by activating the receptor, antagonists in schizophrenia by blocking the receptor or partial agonists used in treating bipolar disorders or addiction [3,4,5].

While all of the listed diseases are connected to the dopaminergic system, they are also characterized by a distinct dysfunctionality of different dopaminergic projection pathways [6]. In PD, the degeneration of dopaminergic neurons in the substantia nigra leads to reduced dopamine levels, thus reducing activation of the D_2_R [4]. In contrast, schizophrenia is defined by hyperproductive, presynaptical dopaminergic neurons in the mesolimbic region, thus overactivating the D_2_R. At the same time, dopaminergic neurons in the prefrontal cortex are hypofunctional, resulting in insufficient activation of the D_1_R due to a lack of dopamine [3]. On the one hand, aberrant signalling involving the D_3_R has been implicated in diseases such as PD, restless leg syndrome and depression, where agonists are used to treat motor dysbalances. On the other hand, D_3_R antagonists have been shown to be useful as antipsychotics [7,8].

For most of those conditions, DR-targeting drugs have been approved by the FDA, successfully ameliorating major symptoms [2]. At the same time, they suffer from major drawbacks due to promiscuous activity at the DR subtypes other than the intended one, as well as closely related GPCRs [9]. Levodopa (L-DOPA), the gold standard in treating PD, successfully reduces the major motoric symptoms, such as bradykinesia and tremor, after biotransformation to dopamine, subsequently activating the D_2_R. Problematically, L-DOPA (and dopamine, respectively) is also known to induce dyskinesia (L-DOPA-induced dyskinesia) due to the promiscuous activation of the D_1_R in long-term treatment conditions [10]. While D_2_R agonists play an important role in treating PD, antagonists act as potent antipsychotics in different psychiatric disorders associated with the DR family. Those antipsychotics are also tightly connected to serious adverse drug events such as extrapyramidal syndrome and neuroleptic malignant syndrome [11,12].

D_2_R-selective drugs are clearly beneficial in treating PD and psychiatric disorders by alleviating the mentioned off-target effects. However, DR-subtype selectivity should not only be seen as a tool to counteract side effects but also to open up novel therapeutic avenues. Selective D_3_R agonists have been shown to be effective in vivo by mitigating the cell death of dopaminergic neurons and improving behavioural performances in mouse models of PD [13,14]. Interesting results have also been obtained in clinical studies establishing pramipexole (a D_3_R-preferring ligand) as an effective dopamine substitute in patients not responding to L-DOPA treatment, simultaneously delaying dyskinesia [15]. Another in vivo study indicated the capability of D_3_R-preferring agonists to reverse motivational deficits related to PD [16]. D_3_R-selective antagonists present a promising opportunity in the treatment of schizophrenia. They appear to be completely devoid of the D_2_R-associated side effects described earlier and also treat negative symptoms, which are not covered by conventional antipsychotics [17,18]. Selective D_1_R agonists are particularly interesting in treating cognitive deficits affecting patients suffering from schizophrenia by targeting the prefrontal cortex. Since the clinical relevance of D_1_R agonists was recognized early on, several selective compounds with diverse chemical scaffolds have been designed throughout the years [19]. While many selective ligands are considered a success, they also suffer from limited oral bioavailability and poor blood–brain barrier (BBB) permeability, thus exposing them to rapid peripheral metabolism. This has mainly been attributed to the presence of catechol functionalities in many of the ligands [20]. Different agonists have shown promising results in improving cognitive impairments and working memory in schizophrenia [21,22,23]. Unfortunately, other studies have provided evidence for D_1_R agonists being responsible for inducing seizures [24,25]. While the seizure-inducing mechanisms and the involvement of structure-activity relationships (SAR) are still not fully understood, the development of novel, potentially non-catechol agonists continues [19,26]. All of these findings clearly indicate the benefits of DR-subtype selective drugs. Moreover, they highlight the necessity of better understanding the molecular mechanisms involved in DR-ligand interactions to rationalize the SARs responsible for specific effects.

Drug research in the field of GPCRs has been benefitting from the ‘golden age of GPCR structural biology’ in the discipline of cheminformatics [27]. Different studies have been utilizing computer-assisted drug-design (CADD) methods to investigate GPCRs and also different DR subtypes [28,29,30]. A particularly interesting study by Bueschbell et al. investigated the selectivity of several known DR ligands (e.g., apomorphine and bromocriptine) with homology modelling and molecular docking approaches [31]. The ever-increasing availability of X-ray or cryo-EM structures of the discussed DR subtypes, D_1_R, D_2_R and D_3_R, aids our ability to comprehend DR ligand selectivity. In total, twelve three-dimensional (3D) protein structures of the D_1_R, five D_2_R structures and three D_3_R structures are accessible in the Protein Data Bank (PDB) database as of March 2023. The advent of cryo-EM technologies enabled the high-resolution depiction of the complex DR subtype structures at ≤3 Å, potentially improving molecular docking approaches that investigate DR ligand selectivity.

The conserved amino acids that create the orthosteric binding pockets (OBPs) of virtually all DR subtypes are well known and described [31,32]. Asp^3.32^ in transmembrane (TM) 3 is responsible for ligand recognition forming a salt bridge with the positively charged amine function of ligands. The serine triade consisting of Ser^5.42^, Ser^5.43^ and Ser^5.46^ positioned in TM5 is important in orienting the respective ligand (especially if a catechol functional group is involved) and considering the ligands’ binding affinity. An aromatic microdomain in TM6 includes Trp^6.48^, Phe^6.51^ and Phe^6.52^ as well as His/Asn^6.55^ and is involved in activating the receptor upon interaction with an agonist. Agonist binding induces the so-called ‘rotamer toggle switch’, a domino-like cascade along TM6 reorienting the named amino acids, eventually triggering receptor activation. Less is known about DR sub-domains or structural elements originating from ligands responsible for selectivity. The D_1_R, although belonging to the D_1_-like DR family, is phylogenetically closest to the β-adrenergic receptors (βARs) [33]. Consequently, it features distinct motifs, responsible for selectivity. A study by Zhuang et al. suggested the involvement of the extracellular loop (ECL) 2, more specifically Ser188, which enables the D_1_R to accommodate bulkier ligands such as SKF81297 and SKF83959 [34]. In comparison, the same ligands would sterically clash with the corresponding amino acid Ile184 in the ECL2 of the D_2_R, consequently resulting in D_1_R-selectivity over the D_2_-like DR family. Considering selectivity between D_2_R and D_3_R, work by Newman et al. revealed a secondary binding pocket (SBP), consisting of multiple amino acids such as Val^2.61^, Leu^2.64^, Phe^3.28^ and conserved Gly and Cys residues located in ECL1 and ECL2, respectively [35]. In more detail, Michino and colleagues have suggested the Gly residue in ECL1 to be the critical selectivity determinant [36]. Additionally, studies have shown that the D_3_R possesses an intrinsically higher affinity towards ligands such as dopamine and quinpirole. Robinson and colleagues have shown that the intracellular loop (ICL) 3 might be responsible for this behaviour. Generating D_2_R hybrids containing the D_3_R–ICL3 motif could increase ligand affinity 10- to 20-fold compared with the wild-type D_2_R. A D_3_R–D_2_R–ICL3 hybrid showed inverse effects [37]. An overview of the described SBP and the different domains involved in DR subtype selectivity is shown in Figure 1.

A great deal of effort has been invested in CADD-approaches to investigate and discover potential DR subtype-selective ligands, thus benefitting drug development in, e.g., neurodegenerative diseases such as PD [28,30,31,34,38,39]. However, due to the complexity of DR selectivity, in silico approaches require in vitro validation. In vitro binding affinities at different DR subtypes can be investigated using, e.g., homogenous time-resolved fluorescence (HTRF) assays, which are standardizable, commercially available and also semi-high-throughput compatible [40,41].

Therefore, the aim of this study was to develop a combined in silico/in vitro approach to assess the selectivity of novel DR ligands at different receptor subtypes using a cell-based HTRF assay as well as a molecular docking approach. Discovering DR-selective ligands as well as providing more detailed insights into their binding behaviour would contribute to better pharmacological tools and new starting points in drug development.

## 2. Materials and Methods

### 2.1. Materials

Bromocriptine mesylate (50 mg; CAYM14598-50) was acquired from avantor VWR (Radnor, PA, USA). A68930 hydrochloride (10 mg; A68930) was acquired from biotechne Tocris (Bristol, United Kingdom). Apomorphine hydrochloride was kindly provided by EVERPharma AT GmbH (Unterach, Austria) within the context of a different project. Compounds tested in vitro (shown in Appendix A) were acquired either from SPECS (https://www.specs.net/, accessed on 30 April 2021) or Maybrige (https://www.thermofisher.com/at/en/home/industrial/pharma-biopharma/drug-discovery-development/screening-compounds-libraries-hit-identification.html, accessed on 17 April 2021). All tested compounds were dissolved in 100 % DMSO (dimethyl sulfoxide, acquired from Sigma-Aldrich, St. Louis, MO, USA) and stored at −80 °C until further use.

### 2.2. Ligand Selection for Combined In Silico/In Vitro Approach

Compounds selected for combined in silico/in vitro investigations were chosen based on two main criteria. First, compounds identified as active D_2_R ligands with the previously developed workflow shown in Zell et al. [42] were selected for further in vitro investigations. Second, other compounds from this study showing normalized decreased fluorescence (NDF) values ≥2-fold increased (during in vitro activity screening, Section 2.10) at any of the investigated DR subtypes compared with the other two DR subtypes were included in further investigations.

### 2.3. Similarity Assessment—Tanimoto Scoring (TS) Matrix

Canonical SMILES codes of all the compounds of interest were imported to Canvas version 3.8 (Canvas, Schrödinger Inc., New York, NY, USA). In Canvas version 3.8, radial fingerprints (Extended Connectivity Fingerprint (ECFP4) [43,44]) of all the molecules (based on 2D structures) were calculated followed by an automated calculation of a TS [45] for each compared pair. TS matrices were exported to Excel (Microsoft, Redmond, WA, USA) as csv files and imported to GraphPad Prism version 8 (GraphPad Software, San Diego, CA, USA) to display heatmaps, color-coding the structural similarities. An increasing coefficient indicated an increasing structural similarity. (Dis-)similarities considering chemical scaffolds were further used to assess observed in silico/in vitro phenomena.

### 2.4. Dataset Assembly for Molecular Docking (ChEMBL Validation)

For the validation of the molecular docking approach, DR ligands with known biological activities were extracted from the ChEMBL (https://www.ebi.ac.uk/chembl/ (accessed on 22 April 2022)) for all three DR subtypes investigated in vitro. Only entries originating from homo sapiens (UniProt accession numbers: D_1_R, P21728; D_2_R, P14416 and D_3_R, P35462) were considered. ChEMBL entries were only selected for further evaluation if (I) Ki values for all three DR subtypes were available for each respective molecule and (II) the in vitro measurements included a valid control. The curated ChEMBL entries were divided into D_1_R-, D_2_R-, D_2_like- and D_3_R-selective subsets. D_1_R-, D_2_R- or D_3_R-selectivity was assumed for molecules with binding affinities ≤1000 nM at the respective subtype and ≥1000 nM at the others. Additionally, the Ki values were required to differ at least by a factor of two. D_2_like-selective compounds showed binding affinities ≤500 nM at D_2_R and D_3_R. The final datasets consisting of 29 (**SC1**–**SC29 [46,47,48,49,50,51,52,53,54,55,56,57,58,59,60]**), 25 (**SC30**–**SC54** [53,61,62,63,64,65,66,67,68,69,70,71,72]), 152 (**SC55**–**SC206** [56,69,72,73,74,75,76,77,78,79,80,81,82,83,84,85,86,87,88,89,90,91,92,93,94,95,96,97,98,99,100,101,102,103,104,105,106]) and 78 (**SC207**–**SC284** [53,56,62,63,65,66,79,87,89,96,99,100,101,103,104,105,106,107,108,109,110,111,112,113,114,115,116,117,118]) molecules, respectively, are shown in Appendix A.

### 2.5. Data Set Preparation for Molecular Docking

All compounds were energetically minimized using the mmFF94 forcefield in OMEGA version 3.0.1.2 prior to molecular docking. (OpenEye Scientific Software, Santa Fe, NM, USA) [119].

### 2.6. Molecular Docking Workflow

Docking was performed using GOLD version 5.8.0 (CCDC, Cambridge, United Kingdom) [120]. Protein structures were not energetically minimized during the docking process. Hydrogens were added to all protein structures. CHEMPLP was used as a scoring function, not allowing early termination. For defining the binding site, all atoms within 6 Å of the bound ligand (depending on the cryo-EM structure) were chosen. The number of GA runs was set to 30.

#### 2.6.1. Molecular Docking—D_1_R

Molecular docking into the D_1_R was performed using the apomorphine-bound cryo-EM structure of the PDB entry 7jvq [34]. Specific settings for the D_1_R structure used during docking are shown in Table 1.

#### 2.6.2. Molecular Dynamics Simulation (MDS)—D_2_R

Details about the MDS are shown in the Appendix A. A detailed description of the MDS calculation is given in Appendix A.

#### 2.6.3. Molecular Docking D_2_R

Molecular docking into the D_2_R ligand binding site was performed using the MDS-modified (see Section 2.6.2) cryo-EM structure of the PDB entry 7jvr [34]. During docking, only ASP114R was specified as flexible (1 rotamer (free)).

#### 2.6.4. Molecular Docking D_3_R

Molecular Docking into the D_3_R was performed using the PD128907-bound cryo-EM structure of the PDB entry 7cmv [121]. Specific settings for the D_3_R structure used during docking are shown in Table 2.

### 2.7. DR Subtypes—BLASTP Alignment

To identify the analogous amino acids of the SBP of D_1_R and D_2_R in respect to D_3_R (defined in [35,36]) a BLASTP alignment was performed (https://blast.ncbi.nlm.nih.gov (accessed on 10 March 2023)) [122]. The relevant amino acids in regard to the SBP of D_3_R are shown in Table 3.

The respective amino acids were used during the in silico selectivity assessment during the validation process and the analysis of the novel DR ligands.

### 2.8. Validation of the Molecular Docking Approach—ChEMBL Dataset(s)

Molecular docking results for each ChEMBL dataset (containing 30 poses for each compound) were uploaded to Pipeline Pilot Client version 9.1 (Accelrys, San Diego, CA, USA) [123]. Duplicates from each molecular docking output were removed. Only top-ranked poses (based on fitness score) of each docked compound were retained in the datasets used for further evaluation. The subsequent docking analysis (top-ranked poses) was performed using DiscoveryStudio (DS) 2018 Client (Accelrys, San Diego, CA, USA). The (modified) DR subtype protein structures were loaded into DS. The conserved Gly residues (shown in Table 3) were marked; centroids were calculated and checked as center of mass (COM). Subsequently, the docked DR subtype-selective output files were loaded into the respective DR protein structure. All molecules were marked and COM was calculated with respect to the Gly residues. Finally, distances between COM (Gly residue) and COM (docked ligands) were calculated in Å.

### 2.9. Docking Analysis—Novel DR Ligands

The docking analysis of the novel ligands was performed using LigandScout version 4.4.4 (Inte:Ligand GmbH, Vienna, Austria). Docked ligands (sd files) were loaded into the different DR protein structures (D_1_R into 7jvq [34]; D_2_R into the MDS-modified D_2_R 7jvr [34]; and D_3_R into 7cmv [121], respectively). All 30 poses of each ligand were individually superimposed and the most frequent pose was assessed visually. Subsequently, DR protein structures as well as molecular docking output files, were loaded in PyMOL (Schrödinger Inc., New York, NY, USA) for each DR subtype individually. The protein including the most frequent respective ligand pose (taking the highest-ranking according to fitness score) was extracted as a pdb file. The resulting pdb files were loaded into DS for calculating distances [Å], as shown in Section 2.8.

### 2.10. HTRF-Based Receptor Binding Studies

All HTRF assays were performed using an HTRF-compatible plate reader (model Tecan Spark (Tecan Group, Männedorf, Switzerland)). The respective settings were specifically modified and optimized for the determination of D_2_R ligand-binding affinities. Binding affinities were determined using the same settings for measurements with D_1_R and D_3_R carrier cells. Experiments were performed using two different emission wavelengths at 620 (control) and 665 (D_2/3_R)/510 (D_1_R) nm, respectively. Fluorophores were excited at 320 nm. A dichroic 510 mirror was used, while lag and integration times of 100 and 400 µs were applied, respectively. Flashes were set to 75. Electronic gain was automatically optimized, while the z-position was optimized based on the well with the highest expected signal. Experiments described in Section 2.11, Section 2.12 and Section 2.14 required the use of two 96-well plates. The first plate was used to determine the gain and the z-position. Subsequently, the determined values were set manually for the second plate to enable direct comparison between the different plates.

### 2.11. Characterization of DR Carrier Cells (D_1_R and D_3_R)—Kd Determination

The cells used for the subsequent screening and detailed investigation of D_1_R and D_3_R ligands were acquired from PerkinElmer/cisbio (Waltham, MA, USA; Tag-lite Dopamine D1 or D3a-labeled Cells, ready-to-use (transformed and labeled), 200 tests; C1TT1D1 and C1TT1D3A, respectively). The cells were stored in liquid nitrogen until further use. Fluorescent-labelled ligands (Dopamine D2 Receptor red antagonist Fluorescent Ligand (L0002RED), stored at −20 °C and Dopamine D1 Receptor green antagonist (L0031GRE), stored at −20 °C), assay buffer (5Xconcentrate Tag-lite Buffer (TLB), 100 mL, stored at +4 °C; LABMED), and 96-well plates (HTRF 96-well low-volume white plate; 66PL96005) required for the in vitro assay were also acquired from PerkinElmer/cisbio. The assay was conducted according to the standard operation protocol (SOP) available from PerkinElmer/cisbio. The 96-well plates were incubated at room temperature for 2 h. The 96-well plates were read as described in Section 2.10. The respective concentrations of the dilution series were performed in triplicates. In total, Kd determination was performed twice.

The characterization of the D_2_R carrier cells is detailed in [42]. 

### 2.12. In Vitro Screening—Assessment of Compound Activity

Materials described in Section 2.1 and Section 2.11 were also used during ligand screening. TLB (1X was prepared diluting 5Xconcentrate TLB in water. For ligand screening, compounds were prepared at a working solution concentration of 40 µM in 1XTLB. Compound **1**, apomorphine, was used as the positive control at the same concentration. The assay was conducted in duplicates following the SOP available from PerkinElmer/cisbio and as described in Section 2.10.

### 2.13. Ligand Selection for Ki Determination

Ligand selection was based on NDF values detailed in Appendix A. Novel D_2_R ligands from our previous study (compounds **2**, **3**, **5**, **6**, **7** and **9**) [42] were selected for selectivity assessment. Additionally, compounds **4**, **8** and **10** were investigated due to an NDF fold-difference ≥2 of any of the three DR subtypes compared with the other two.

### 2.14. K_I_ Determination for Selected Ligands

The materials described in 2.1 and 2.11 were also used for Ki determination of the ligands selected after screening. The selected ligands (compounds **1**–**10**) were diluted in 1x TLB. Compounds **1**, **2** and **4**–**9** were diluted to an initial working solution concentration of 4 × 10^−4^ M. Compounds **3** and **10** were diluted to an initial working solution concentration of 1 × 10^−4^ M. Different concentrations were chosen due to differences in aqueous solubility of the compounds. The Ki was determined in duplicates following the SOP available from PerkinElmer/cisbio and as described in Section 2.10. 

### 2.15. Data Processing, Representation and Analysis

Saturation binding curves were processed and visualized in GraphPad Prism 8 (Nonlinear regression (curve fit), One site—Fit logIC_50_ was performed using GraphPad Prism version 8.2.1 for Windows, GraphPad Software, San Diego. CA, USA). Molecular docking was performed in GOLD 5.8.0 (CCDC, Cambridge, United Kingdom) [120]. Docking analysis was performed in LigandScout version 4.4.5 (Inte:Ligand GmbH, Vienna, Austria) [124]. MDS and calculations for distance-based in silico approach were performed in DS Client 2018 (DiscoveryStudio, Accelrys Inc., San Diego, CA, USA). Two-dimensional structures of all shown compounds were generated using ChemDraw version 19.0 (PerkinElmer, Waltham, MA, USA). SD files used for similarity assessment were generated using PipelinePilot Client 9.1 (Dassault Systems, BIOVIA Discovery Studio, San Diego, CA, USA, 2018). Similarity assessment was performed using Canvas 3.8 (Canvas, Schrödinger, LLC, New York, NY, USA, 2021). Docking alignments and visualization were performed in PyMOL (PyMOL, Schrödinger, LLC, New York, NY, USA, 2021).

## 3. Results

### 3.1. Structural Summary of the Investigated Ligands

All compounds investigated in silico and in vitro within this study are shown in Figure 2.

The novel ligands (compounds **2**–**10**) investigated with the combined approach within the scope of this study were structurally compared with each other using a Tanimoto scoring (TS) matrix. Therefore, the observed in silico and/or in vitro phenomena could be potentially correlated to structural (dis-)similarities. The TS matrix is shown in Figure 3, ranging from 0 (green) to 1 (red), corresponding to structurally unrelated and identical compounds, respectively.

Thirty-three out of thirty-six pairs scored between 0.03 and 0.15, thus representing a structurally diverse compound collection. Only three pairs, i.e., compounds **5** and **6** (TS 0.28), **6** and **10** (TS 0.21) and **9** and **10** (TS 0.21), were characterized by a similarity score of >0.21, reflecting a higher degree of similarity (considering the use of radial fingerprints).

### 3.2. In Vitro Compound Screening—An Assessment of DR Subtype Selectivity

The investigated compounds were taken from a previous pharmacophore-based virtual screening study described in Zell et al. [42]. All 2D structures (compounds **2**–**10** and **SC285**–**SC365**) and respective NDF values for all DR subtypes are shown in Appendix A. The activities of all compounds were investigated via a competitive binding (in comparison with a fluorescence-labeled ligand) of the respective compounds at D_1_R/D_3_R, utilizing an HTRF assay using a screening concentration of 10 µM. NDF values of compounds chosen for further evaluation are shown in Table 4.

Based on the resulting NDF values, all compounds but **4** and **8** showed promiscuous receptor activities suggesting diverse selectivity profiles. Only compounds **4** and **8** showed NDF values close to 1 at both D_1_R and D_2_R, suggesting inactivity at those DR subtypes and, respectively, selectivity for the D_3_R.

### 3.3. Ki Determination—Of the Selected Compounds at DR Subtypes

The selected compounds were investigated in vitro to determine their binding affinities (Ki values) at the three different DR subtypes, D_1_R, D_2_R and D_3_R. In Figure 4a,b, compounds **2** and **10** are shown as examples. Compound **2** represents a non-selective ligand while compound **10** is characterized by the highest selectivity (for D_3_R). The remaining binding curves are shown in Appendix A.

The Ki values of all investigated ligands, as well as the calculated fold-differences for each receptor pair, are shown in Table 5.

All the compounds investigated in vitro, except compound **1**, showed a clear D_2_like selectivity with preferences for D_3_R (D_1_R/D_3_R fold differences ranging from 3.06 to 1031.4, D_2_R/D_3_R fold-differences ranging from 1.66 to 263.7, respectively). While compounds **2**, **5**, **6**, **9** and **10** showed higher affinities for D_1_R compared with the D_2_like DR subtypes, the affinities of compounds **3**, **4** and **8** were not determinable for D_1_R, and were thus considered inactive. Compounds **4** and **8** were also inactive at D_2_R, and were thus considered D_3_R-selective. Only compound **1** was characterized by the lowest binding affinity for D_2_R. Interestingly, all compounds showed the highest affinity at the D_3_R.

### 3.4. Dataset Assembly—In Silico Assessment

To validate the molecular docking approach utilized to assess compound selectivity in silico, DR ligands with different selectivities for the DR subtypes D_1_R, D_2_R and D_3_R with known biological activities were extracted from the ChEMBL database. Compounds were only included in the final datasets if (I) their binding affinities were determined in vitro at all three DR subtypes and (II) their in vitro measurements included a valid control to assess assay functionality. The curated ChEMBL entries were divided into D_1_R-, D_2_R-, D_2_like- and D_3_R-selective subsets. D_1_R-, D_2_R- or D_3_R-selectivity was assumed for molecules with binding affinities ≤1000 nM at the respective subtype and ≥1000 nM at the others. D_2_like-selective compounds showed binding affinities ≤500 nM at D_2_R and D_3_R. The final datasets consisting of 29 (**SC1**–**SC29** [46,47,48,49,50,51,52,53,54,55,56,57,58,59,60]), 25 (**SC30**–**SC54** [53,61,62,63,64,65,66,67,68,69,70,71,72]), 152 (**SC55**–**SC206** [56,69,72,73,74,75,76,77,78,79,80,81,82,83,84,85,86,87,88,89,90,91,92,93,94,95,96,97,98,99,100,101,102,103,104,105,106]) and 78 (**SC207**–**SC284** [53,56,62,63,65,66,79,87,89,96,99,100,101,103,104,105,106,107,108,109,110,111,112,113,114,115,116,117,118]) molecules, respectively, are shown in Appendix A.

### 3.5. Validation of Molecular Docking

The utilized molecular docking approach was based on the work of Michino and colleagues [36]. Therefore, different ChEMBL datasets, previously defined as DR subtype-selective (see Section 3.4), were docked into the 3D protein structures of D_1_R, D_2_R and D_3_R (molecular docking workflow described in Section 2.6.1, Section 2.6.3 and Section 2.6.4). Due to the high number of investigated ligands (>300), only the top-ranked poses (considering the fitness score) were considered during further analysis. The COM for all ligands included in each specific DR-selective subset was calculated using DS. Distances of each COM with respect to each DRs conserved Gly residue (shown in Table 3) were calculated in [Å]. The calculated fold-differences for each subset, comparing different DRs with each other, are shown in Figure 5 (absolute distances determined in DS are given in Appendix A).

The dashed red line shown in Figure 6 indicates a fold-difference of 1.0, which indicates the same distance between the ligands collective COM and the conserved Gly residue after docking into the respective DR structures. All investigated datasets show a fold-difference close to 1.0 considering the D_2_R/D_3_R comparison. This means that they showed an almost identical distance between COM and the Gly residue. In contrast, all datasets showed an increased fold-difference >1.0, when comparing D_1_R with D_2_R or D_3_R, respectively. Details considering all datasets are shown in Table 6.

Clearly, the approach was incapable of distinguishing D_2_R- and D_3_R-selectivity from each other based on the COM–Gly distance. However, the utilized molecular docking approach was capable of identifying D_2_like-selective ligands based on their position within the respective DRs OBP. 

### 3.6. In Silico Assessment of DR Selectivity—Interaction with the SBP

For the in silico assessment of the selected compounds **1**–**10**, the most frequent poses after docking were used. After calculating the fold-differences based on the distances between each ligand’s individual COM and the respective Gly residue (within each of the three DR subtypes SBP, shown in Appendix A), they were plotted against the fold-differences based on the DR-specific Ki values (shown in Table 5) determined in vitro. The resulting scatter plot is shown in Figure 6.

In addition to the individual data points, Figure 6 shows regression curves for all DR pairs. The D_1_R/D_2_R curve (dots) was characterized by the steepest slope suggesting the capability of the in silico approach in discriminating D_2_R-selective ligands. While the slope for the D_1_R/D_3_R curve (squares) was less steep, the calculated fold-differences (based on distance, *y*-axis) was already higher at lower Ki-based fold-differences (*x*-axis), indicating a similar capability to discriminate D_3_R-selective ligands. The D_2_R/D_3_R curve (triangles) was flatter, with individual values scattered around 1.0. Consequently, this reflected the results shown in Figure 5, where D_2_R/D_3_R-selectivity could not be discriminated based on the selected approach. In summary, the developed distance-based in silico approach was highly capable in identifying D_2_like-selectivity. This was also indicated by the R^2^ values (shown in Figure 6) regarding the D_1_R/D_2_R and the D_1_R/D_3_R comparison showing a positive correlation between increasing binding affinities with increasing selectivity.

### 3.7. Retrospective Analyis of the In Silico/In Vitro Correlation

To get a more detailed insight into the binding mode of each of the investigated ligands at the respective DR subtype, the most frequent docking poses of each compound (Figure 7, Figure 8 and Figure 9 and Appendix A) were visualized in the different binding pockets using PyMOL. Figure 7 and Figure 8 show the different binding poses of the non-selective compound **2** and compound **10**, which had the highest D_3_R-selectivity.

In Figure 7, the tertiary amine functionality (contained in the piperazine motif) of compound **2** is clearly oriented towards the OBP, allowing the formation of the salt-bridge with Asp^3.32^ (described as the crucial interaction to define a DR ligand). While the position of compound **2** was flipped in D_2_R in comparison with D_1_R and D_3_R (highlighted by the orientation of the chlorine, green), the overall positioning of compound **2** was similar in each DR subtype. Consequently, there was no distinct orientation of any of the poses towards the SBP, resulting in the non-selective binding with Ki fold-differences between 1.2 and 2.2 (see Table 5).

In Figure 8, the tertiary amine functionality (contained in the piperazine motif) of compound **10** was again oriented towards the OBP. Thus, the salt-bridge formation with Asp^3.32^ was possible. In contrast to compound **2**, the binding poses of compound **10** were distinctly different in the respective DR subtypes. Comparing the poses in D_1_R and D_3_R, the D_3_R pose (green) was shifted slightly to the right towards the SBP. The D_2_R binding pose (orange) was clearly different from both D_1_R and D_3_R, with the chloro-substituted ring clearly oriented towards the SBP. While this explained the observed D_1_R/D_2_R fold difference of 3.7, it did not correlate with the D_2_R/D_3_R fold-difference of 331.8. However, the detailed analysis of the binding poses of compound **10** correlated with the observed D_2_like selectivity determined in vitro. Additionally, it also partially confirmed the retrospective results of the distance-based approach shown Figure 6, highlighting the capability of the developed approach to identify D_2_like-selectivity.

These findings were also supported by the in-depth analyses of compounds **3**, **5**, **6**, **7** and **9** (PyMOL alignments shown in Appendix A), where D_2_R and D_3_R poses were distinctly oriented towards the SBP. However, in agreement with the findings considering compound **10**, the D_2_R- and D_3_R binding poses did not correlate with the higher D_3_R binding affinities found in vitro. Again, the results allowed for the confirmation of D_2_like-selectivity of the investigated ligands.

Compounds **4** (Figure 9) and **8** (Appendix A) were the only compounds with no determinable binding affinity at D_1_R and D_2_R, additionally showing slightly increased distance-based fold-differences (Figure 6 and Appendix A), regarding D_2_R/D_3_R-selectivity, of 1.10 and 1.04, respectively.

This was also reflected in the binding pose of compound **4**, where the D_3_R pose (green) was oriented closer to the SBP. Similar results were observed in the binding pocket comparison of compound **8**.

## 4. Discussion

The characterized DR ligands showed different selectivity profiles. Interestingly, all ten compounds investigated by the developed in silico/in vitro approach (including the novel compounds **2**–**10**) showed either D_3_R-preferences or clear D_3_R-selectivity. Compound **2**, for example, showed fold-differences of 3.23 and 1.66 for D_1_R/D_3_R and D_2_R/D_3_R, respectively, thus exerting D_3_R-preferences. Compounds **4** and **8** were characterized by no determinable binding affinities at D_1_R and D_2_R, consequently they were categorized as D_3_R-selective. While compound **10** showed low to intermediate binding affinities at D_1_R (2.38 µM) and D_2_R (0.61 µM), it also exerted the highest quantifiable selectivity fold-differences with values of 1031.4 and 263.7 for D_1_R/D_3_R and D_2_R/D_3_R, respectively. Additionally, all investigated compounds but **1** were D_2_like-selective.

The rather promiscuous behavior of compound **2** is attributed to its structural similarity to clozapine, the prototypical representative of atypical antipsychotics (a drug class belonging to the atypical antipsychotics). While clozapine is characterized by its potent antipsychotic effect, it is also known as a so-called ‘dirty drug’ due to its promiscuous activity at a variety of aminergic GPCRs (including dopaminergic, serotonergic and adrenergic receptor families) [125]. Thus, a similar pharmacological profile of compound **2** was expected. This was not only confirmed by the in vitro data but also by the developed in silico approach, correlating the positioning of the ligand within the OBP and SBP with its respective DR subtype selectivity. Even though the binding behavior of compound **2** appeared non-selective, the in silico approach was capable of detecting the slight D_2_like-preference resulting in distance-based fold-differences of 1.33 and 1.28 for D_1_R/D_2_R and D_1_R/D_3_R, respectively. Moreover, the compound could be active at other GPCRs which were not investigated within this study. In general, the investigated compounds could be biologically active at other PCRs. For example, a study by Garcia-Romero and colleagues identified several antiparkinsonian molecules with polypharmacological profiles. Those molecules were also biologically active at other GPCRs such as muscarinic acetyl choline receptors and adenosine receptors but also at the norepinephrine transporter [126]. This study highlights the necessity of investigating the identified ligands in even more detail to potentially exploit potential polypharmacological aspects and, even more importantly, to identify possible off-target activities. However, this study deliberately focused on isolated ligand–receptor interactions to generate reliable in silico/in vitro correlation, thus elaborating upon DR selectivity mechanisms.

The comparison of compounds **6** and **10** allowed for very interesting insights into the DR subtype-selectivity profile of structurally similar ligands differing mainly regarding linker lengths. Compounds **6** and **10** are both characterized by two terminal aromatic rings and a linker region consisting of a piperazine motif, an amide functionality and an alkyl chain (see Figure 2). Moreover, they also share binding preferences at the different DR subtypes following D_1_R > D_2_R > D_3_R. Both compounds showed comparable Ki-based fold-differences of 5.77 (compound **6**) and 3.91 (compound **10**) for D_1_R/D_2_R. However, the D_1_R/D_3_R and D_2_R/D_3_R fold-differences increased drastically for compound **10** (1031.4 and 263.7, respectively) compared with compound **6** (20.7 and 3.59, respectively). Michino and colleagues showed similar phenomena in their study investigating the impact of the linker length in analogues of the highly D_3_R-selective compound R22 ([(R)-N-(4-(4-(2,3-dichlorophenyl)piperazin-1-yl)-3-hydroxybutyl)-1H-indole-2-carboxamide]) [36,79]. The investigated R-22 analogues included C3- to C5-linker regions. The C3-linker length resulted in non-selective binding behavior at D_2_R and D_3_R. The C5-linker length markedly reduced D_2_R/D_3_R selectivity. Only the C4 analogue retained a significant D_2_R/D_3_R-selectivity with a 45.7 fold-difference. Compound **6**, including a C2-linker region, showed a comparably reduced D_2_R/D_3_R-selectivity of 3.59. In contrast, compound **10**, including a C4-linker region, exerted a fold-difference of 263.7. While compounds **6** and **10** are only partially related (similarities shown in red) to the R22-analogues (see Figure 10), the observed in vitro effects are potentially attributable to the length of the linker region.

Compounds **2**, **3**, **5**, **6**, **7** and **9** were already reviewed in our earlier publication investigating their novelty, as was the DR-associated effects of their closest structural relatives [42]. While none of the investigated structures yielded exact structural matches, the most similar structures were associated with different DR-related effects. Structurally similar compounds to **5** and **6** were associated with D_4_R-selectivity but no defined mode of action (agonism or antagonism) [127]. A compound similar to **2** was associated with D_4_R antagonism, while structurally similar ligands regarding compounds **3** and **7** were investigated considering D_2_R antagonism [128,129,130,131]. Only a compound structurally similar to **9** was associated with D_3_R-selectivity and D_2_R antagonism [132]. The novel ligands included within this study were compared with the literature using SwissTargetPrediction (http://www.swisstargetprediction.ch/ (accessed on 5 March 2023)) and SwissSimilarity (http://www.swisssimilarity.ch/ (accessed on 5 March 2023)) [133,134,135]. Compounds **4** and **8** yielded low scores in SwissTargetPrediction where the identified similar compounds (ChEMBL IDs 59603 and 592377) had been investigated considering D_1_R- and D_2_R activity but not D_3_R selectivity [136,137]. ChEMBL entry 4081151 was structurally closely related to compound **4** but had only been investigated for kappa opioid receptors [138]. A SwissSimilarity match for compound **8** (ChEMBL ID 1094101) was investigated for its binding affinity at serotonergic receptors and aminergic GPCR family members, but not in respect to DRs [139]. Thus, compounds **4** and **8** open up novel insights into D_3_R-selectivity. Compound **10** resulted in exact structural matches and closely related matches in both SwissTargetPredicition and SwissSimilarity investigating D_3_R-selectivity. Still, the comparison between compounds **6** and **10** contributes to a better understanding of the role of the linker length on the DR subtype selectivity of structurally related, but not identical, chemical scaffolds.

As mentioned earlier, all novel ligands exerted their highest binding affinities at the D_3_R. This is partially in accordance with the scientific literature, where the D_3_R shows a high intrinsic binding affinity for agonists such as dopamine (420-fold increased affinity) and quinpirole [37,140]. While this is attributed to intracellular loop 3 in D_3_R, this characteristic has only been shown for agonists. However, the known characteristics of the structurally related compounds of the novel compounds described above suggest a low probability that all investigated ligands are actually agonists. Thus, the increased D_3_R affinity of compounds **2**–**10** presumably originates from a distinct interaction with the described SBP [36]. The developed in silico approach proposes a workflow to identify D_2_like-selectivity. However, the static nature of the molecular docking approach does not allow for discrimination of D_2_R/D_3_R-selectivity. This limitation can be attributed to the very dynamic nature of the EL structural motifs of the D_2_like DRs responsible for subtype selectivity. Different studies propose MDS approaches to circumvent the shortcomings of molecular docking approaches and to account for protein flexibility [36,141].

Thus, the developed in silico/in vitro workflow clearly demonstrated its potential use in preclinical drug research by enabling the identification of D_2_like-selective ligands independently of chemical scaffolds. This could be especially important in diseases of the CNS, where D_1_R activation has been associated with induction of seizures. In addition, the D_3_R is a fast emerging molecular target of interest in treating PD. Thus, the accurate prediction of D_2_like-selectivity could act as an important starting point in developing truly D_3_R-selective compounds and also providing pharmacological tools to aid in the understanding of D_2_like DR subtype selectivity.

## 5. Conclusions

In this study, ten compounds were investigated for their DR subtype selectivity. A combined in silico/in vitro approach was developed to correlate the positioning within the receptor binding pocket with the biological activity. With the workflow, we observed a correlation between the distance of the ligand to the conserved glycine residue within the secondary binding pocket and the DR subtype selectivity. Most prominently, the workflow was able to identify D_2_like-selectivity but could not explain D_2_R/D_3_R selectivity observed in vitro. The most selective compound, **10,** was characterized by a low nano-molar activity at D_3_R (Ki = 2.3 nM) showing a distinct selectivity over D_1_R and D_2_R with fold-differences of 1031.4 and 263.7, respectively. This study provides a valuable tool in further understanding DR subtype selectivity mechanisms, thus aiding the development of more selective DR ligands.

## Figures and Tables

**Figure 1 biomedicines-11-01468-f001:**
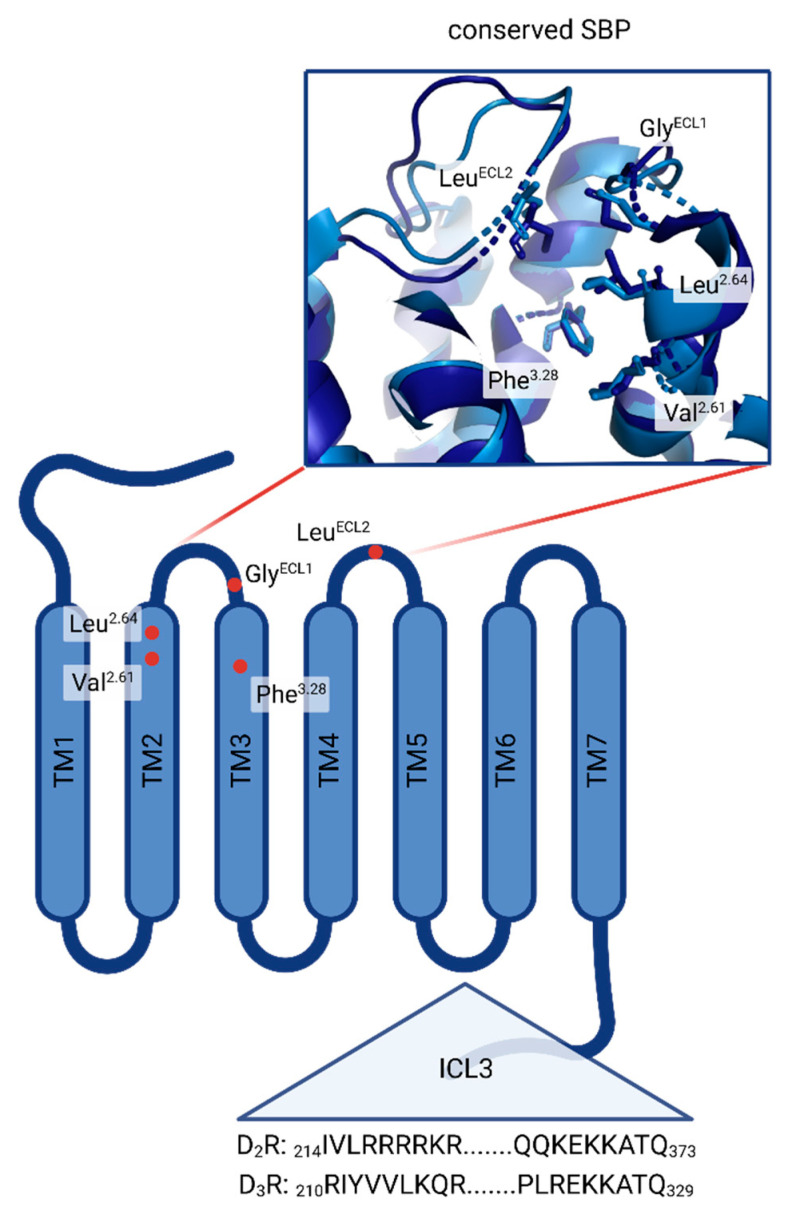
Overview of DR sub-domains relevant in DR subtype selectivity. Red dots highlight the highly conserved amino acids Val^2.61^, Leu^2.64^, Phe^3.28^, Gly^ECL1^ and Leu^ECL2^ in the SBP of D_2_R and D_3_R. Zoomed in box of the conserved SBP shows the 3D arrangement. Partial primary sequences (amino acid positions are shown in the index) of ICL3 are shown for both D_2_R and D_3_R. (Created with BioRender.com (accessed on 13 April 2023)).

**Figure 2 biomedicines-11-01468-f002:**
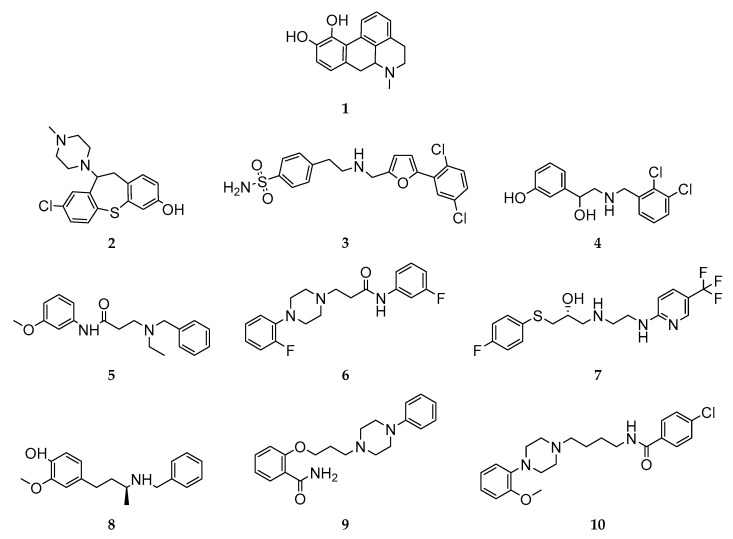
Overview of the 2D structures of ligands investigated in silico and in vitro.

**Figure 3 biomedicines-11-01468-f003:**
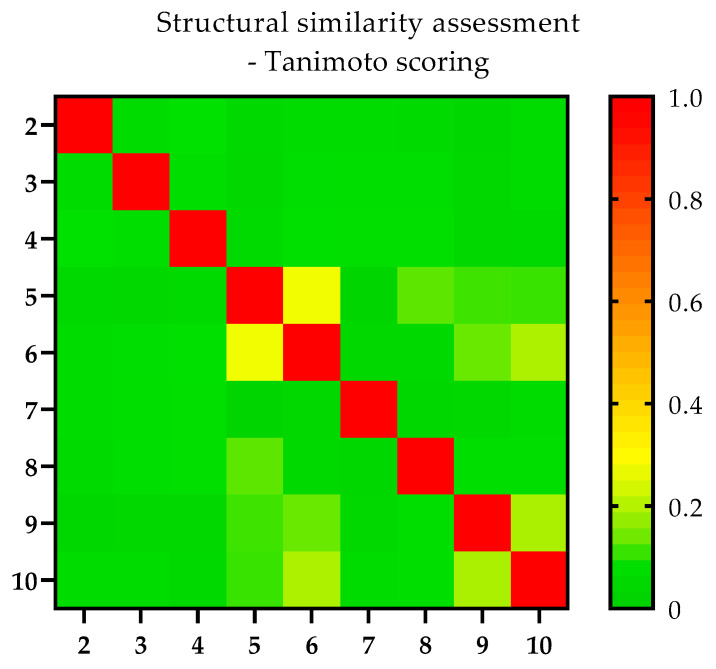
Overview of the structural (dis-)similarity of the investigated novel ligands (compounds **2**–**10**). (Dis-)similarity was assessed utilizing a TS matrix based on radial fingerprints (ECFP4). TS ranging from 0 (green) to 1 (red) showing unrelated and identical structures, respectively. Numbers given on the *x*- and *y*-axis indicate the different compounds investigated within this study.

**Figure 4 biomedicines-11-01468-f004:**
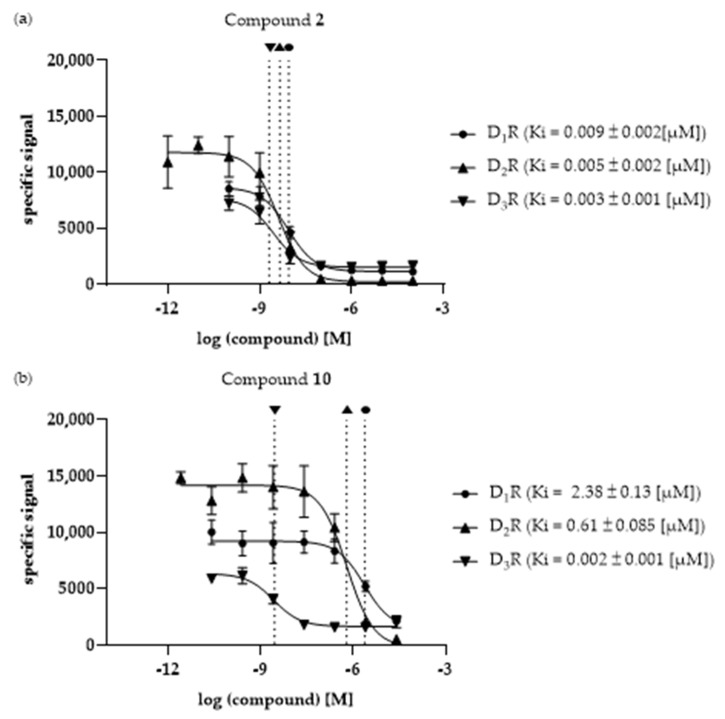
Comparison of the Ki values of compounds (**a**) **2** and (**b**) **10** determined at the investigated DR subtypes D_1_R, D_2_R and D_3_R. Vertical, dotted lines indicate the respective Ki values at the different DR subtypes. Ki values [µM] ± SD were determined with *n* = 6.

**Figure 5 biomedicines-11-01468-f005:**
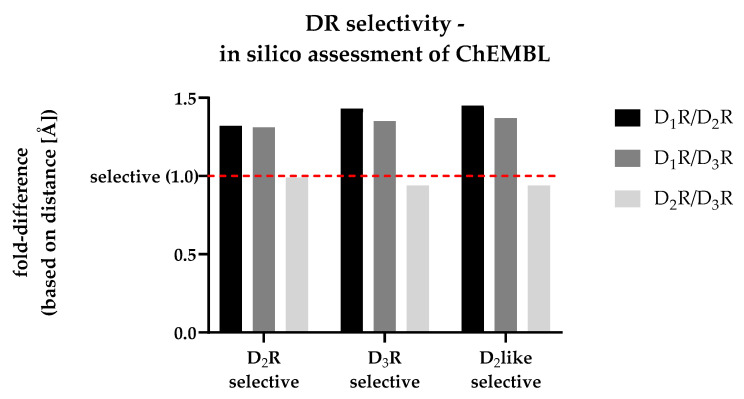
Comparison of the fold-differences based on the distances [Å] between each DR-selective subsets COM and the respective conserved Gly residue. The dashed red line shows a distance-based fold-difference of 1.0, indicating a non-selective profile of the respective dataset based on the in silico analysis.

**Figure 6 biomedicines-11-01468-f006:**
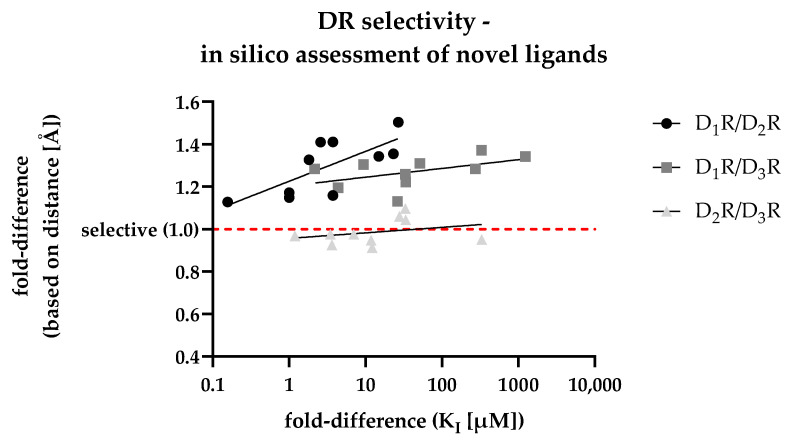
Correlation of in vitro determined fold-differences (*x*-axis) and in silico determined distance-based fold-differences (*y*-axis) to investigate DR subtype selectivity of novel compounds. The dashed red line shows a distance-based fold-difference of 1.0, indicating a non-selective profile of the respective compound based on the in silico analysis.

**Figure 7 biomedicines-11-01468-f007:**
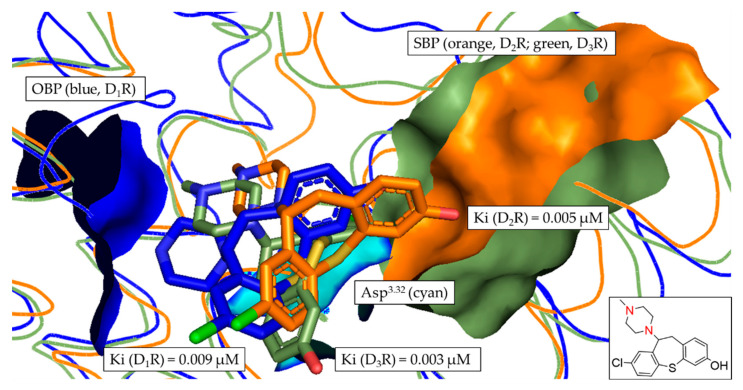
Alignment of the most frequent poses of compound **2** docked into D_1_R, D_2_R and D_3_R. The surface of the highly conserved OBP is shown in blue (based on D_1_R) consisting of Asp^3.32^ (highlighted in cyan) and Ser^5.42/5.43/5.46^. The conserved SBP-surface is displayed in orange (D_2_R) and green (D_3_R) consisting of Val^2.61^, Leu^2.64^, Gly^EL1^, Phe^3.28^ and Cys^EL2^ (individual amino acid labels shown in Table 3), respectively. Ki values determined in vitro are shown for each DR subtype. Two-dimensional structure of compound **2** is shown. Amine functional group involved in formation of the salt-bridge is highlighted in red.

**Figure 8 biomedicines-11-01468-f008:**
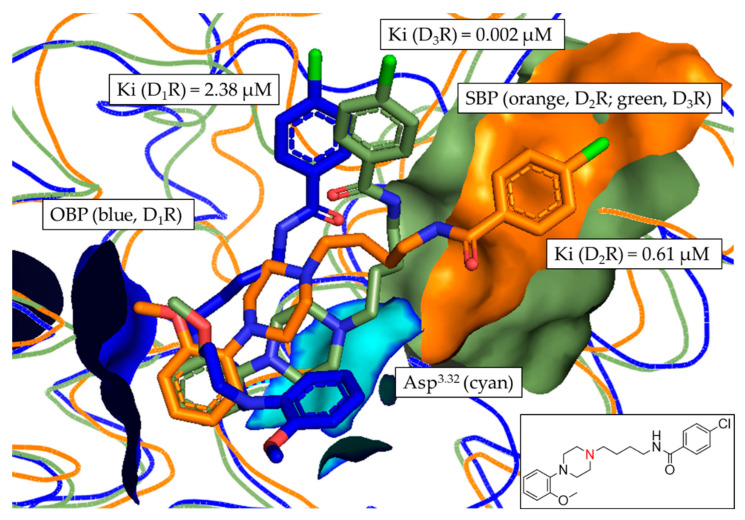
Alignment of the most frequent poses of compound **10** docked into D_1_R, D_2_R and D_3_R. The surface of the highly conserved OBP is shown in blue (based on D_1_R) consisting of Asp^3.32^ (highlighted in cyan) and Ser^5.42/5.43/5.46^. The conserved SBP-surface is displayed in orange (D_2_R) and green (D_3_R) consisting of Val^2.61^, Leu^2.64^, Gly^EL1^, Phe^3.28^ and Cys^EL2^ (individual amino acid labels shown in Table 3), respectively. Ki values determined in vitro are shown for each DR subtype. Two-dimensional structure of compound **10** is shown. Amine functional group involved in formation of the salt-bridge is highlighted in red.

**Figure 9 biomedicines-11-01468-f009:**
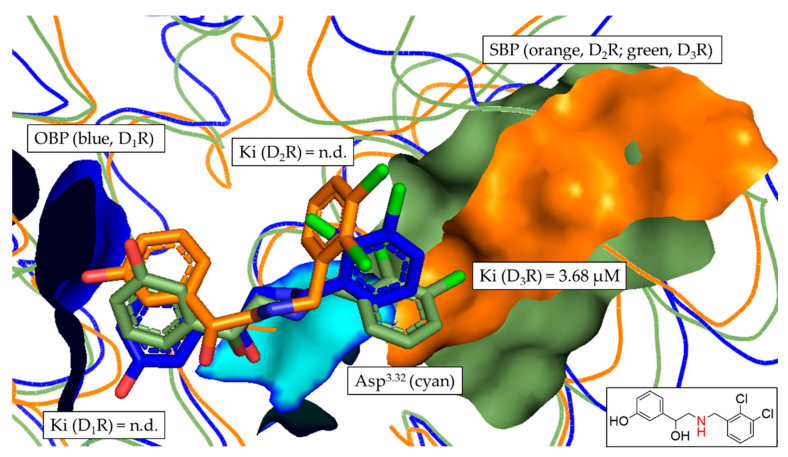
Alignment of the most frequent poses of compound **4** docked into D_1_R, D_2_R and D_3_R. The surface of the highly conserved OBP is shown in blue (based on D_1_R) consisting of Asp^3.32^ (highlighted in cyan) and Ser^5.42/5.43/5.46^. The conserved SBP-surface is displayed in orange (D_2_R) and green (D_3_R) consisting of Val^2.61^, Leu^2.64^, Gly^EL1^, Phe^3.28^ and Cys^EL2^ (individual amino acid labels shown in Table 3), respectively. Ki values determined in vitro are shown for each DR subtype. Two-dimensional structure of compound **4** is shown. Amine functional group involved in formation of the salt-bridge is highlighted in red. n.d., not determinable.

**Figure 10 biomedicines-11-01468-f010:**
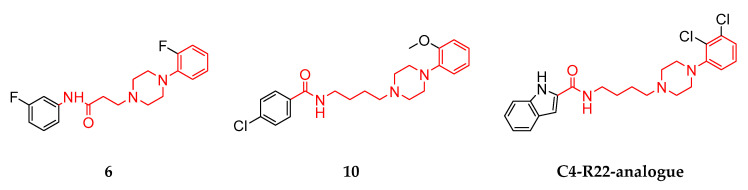
Comparison of the chemical scaffolds of compounds **6**, **10** and the R22-analogue. Structural elements highlighted in red show similarities between the different compounds also indicating the differences in linker length.

**Table 1 biomedicines-11-01468-t001:** Summary of the amino acid flexibility settings of the D_1_R cryo-EM structure used during docking.

Setting	Value
Flexible Sidechains	ASP103 R, 1 rotamer (free)
	TRP285 R, 1 rotamer (free)
	PHE288 R, 1 rotamer (free)
	PHE289, 1 rotamer (free)
	ASN292 R, 1 rotamer (free)

**Table 2 biomedicines-11-01468-t002:** Summary of the amino acid flexibility settings of the D_3_R cryo-EM structure used during docking.

Setting	Value
Flexible Sidechains	ASP110 R, 1 rotamer (free)
	HIS349 R, 8 rotamers (constrained)

**Table 3 biomedicines-11-01468-t003:** Overview of the amino acids forming the SBP in different DR subtypes. D_2_like subtypes include D_2_R and D_3_R.

DR Subtype			
D_3_R	D_1_R	D_2_R	Status
Val86	Lys81	Val91	Conserved in D_2_like DRs
Leu89	Ala84	Leu94	Conserved in D_2_like DRs
Gly94	Gly88	Gly98	Conserved
Phe106	Trp99	Phe110	Conserved in D_2_like DRs
Cys181	Cys186	Cys182	Conserved

**Table 4 biomedicines-11-01468-t004:** Summary of the in vitro screening of known and potential DR ligands considered selective for one of the three investigated subtypes. All measurements were conducted at a concentration of 10 µM (*n* = 4). Fluorescence decrease was normalized to the control. Cpd., compound.

Cpd.	Normalized Decrease in Fluorescence (NDF) ± SD		
	D_1_R	D_2_R ^a^	D_3_R
Control	1	1	1
**1**	3.47 ± 1.04	9.44 ± 5.97	3.82 ± 1.32
**2**	5.46 ± 1.97	40.41 ± 1.39	4.16 ± 1.08
**3**	1.78 ± 1.35	3.99 ± 2.58	3.96 ± 1.10
**4**	0.90 ± 0.31	0.90 ± 0.39	2.75 ± 0.61
**5**	1.26 ± 0.64	15.74 ± 18.15	2.65 ± 0.64
**6**	2.15 ± 1.16	8.18 ± 3.62	4.21 ± 0.79
**7**	1.57 ± 0.54	10.85 ± 4.93	3.79 ± 0.70
**8**	1.20 ± 0.59	1.10 ± 0.52	2.63 ± 0.52
**9**	1.71 ± 0.86	22.08 ± 6.62	3.97 ± 0.78
**10**	2.59 ± 1.04	22.89 ± 8.41	4.09 ± 1.07

^a^ Values were extracted from previous experiments detailed in [42].

**Table 5 biomedicines-11-01468-t005:** Summary of the determined Ki values of all ligands investigated in vitro. Binding affinities are shown for the three different DR subtypes D_1_R, D_2_R and D_3_R. Ki values were determined using *n* = 6. Apomorphine (**1**) was used as a control. Cpd., compound.

Cpd.	Ki [µM]			Selectivity		
	D_1_R	D_2_R	D_3_R	D_1_R/ D_2_R	D_1_R/ D_3_R	D_2_R/ D_3_R
**1**	0.36 ± 0.009	2.36 ± 0.14	0.12 ± 0.048	0.15	3.06	19.8
**2**	0.009 ± 0.002	0.005 ± 0.002 ^a^	0.003 ± 0.001	1.95	3.23	1.66
**3**	n.d. ^b^	4.66 ± 2.69 ^a^	0.38 ± 0.022	>21.4 ^b^	262.2	12.2
**4**	n.d. ^b^	n.d. ^b^	3.68 ± 0.94	-	>27.2 ^b^	>27.2 ^b^
**5**	46.9 ± 27.4	10.95 ± 4.43 ^a^	2.25 ± 0.91	4.28	20.8	4.86
**6**	7.76 ± 4.41	1.35 ± 0.63 ^a^	0.37 ± 0.28	5.77	20.7	3.56
**7**	8.33 ± 2.17	2.78 ± 1.06 ^a^	0.68 ± 0.068	3.00	12.3	4.11
**8**	n.d. ^b^	n.d. ^b^	2.32 ± 0.92	-	>43.1 ^b^	>43.1 ^b^
**9**	9.46 ± 1.18	0.33 ± 0.093 ^a^	0.024 ± 0.003	28.6	395.1	13.8
**10**	2.38 ± 0.13	0.61 ± 0.085	0.002 ± 0.001	3.91	1031.4	263.7

^a^ Values were extracted from previous experiments detailed in [42]. ^b^ Ki values could not be quantitatively determined. To calculate values for selectivity, K_I_ values were assumed to be ≥100 µM (highest concentration used during in vitro testing). n.d., not determinable.

**Table 6 biomedicines-11-01468-t006:** Summary of the distance-based docking approach of different DR-subtype selective ChEMBL datasets. Calculated fold-differences were based on distances [Å] between COM and the respective conserved Gly residue.

Dataset	Fold-Difference (Cons. Gly-COM))		
	D_1_R/D_2_R	D_1_R/D_3_R	D_2_R/D_3_R
D_2_R selective	1.32	1.31	0.99
D_3_R selective	1.43	1.35	0.94
D_2_like selective	1.45	1.37	0.94

## Data Availability

The data presented in this study are available in the Appendix A. If further data are required, they are available from the corresponding author upon request.

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
