# Peer review of "Dopamine Receptor Ligand Selectivity—An In Silico/In Vitro Insight"

_biomedicines, 2023, doi:10.3390/biomedicines11051468_

Round 1

Reviewer 1 Report

Manuscript Number: Biomedicines-2374148

 Full Title: Dopamine Receptor Ligand Selectivity - An in silico / in vitro Insight

The main objective of this work is the in silico / in vitro evaluation of potential candidates as selective dopamine receptor ligands.

The work, in general, is well written, of easy lecture and interesting, but I would like to make the following considerations to the authors:

-      In silico and in vitro are Latin terms, so I would write them in italics.

 -      The acronym for dopamine receptors (DA) can be DR or DAR, for me, DAR is more correct. But I assume that it is something debatable.

 -      Even if they are known, it is best to use the full description before using the acronym. For example, for FDA, SBP, ECFP4, TLB, NDF, and more.

 -      On the first two pages there are four paragraphs beginning with "However”. One suggestion is that the first "However" can be changed to "Moreover". Evaluate other possible changes.

 -      What percentage of DMSO was used to dissolve the compounds?

 -      Some company names and/or addresses are missing. For example, who is VMR and what is his address? Tocris is a proper name, use a T; what is your address? The address of EVERPharma AT GmbH?

 -      In section 2.9 use subscripts on dopamine receptors.

 -      For me, the most correct abbreviation is Ki (small i)

 -      Clozapine is not the representative of tricyclic antipsychotics. It is Chlorpromazine, compound 6.6.6, it was the first synthetic drug used as an antipsychotic.

Clozapine is the prototype of the atypical antipsychotics, and all other atypical drugs are called clozapine-like.

Author Response

We would like to thank reviewer 1 very much for the constructive feedback and the very thorough read-through of our manuscript. The comments helped us to further improve the quality of our manuscript. In the following we list the comments of reviewer 1 and answer to them accordingly.

In silico and in vitro are Latin terms, so I would write them in italics.

The reviewer is absolutely correct stating that both terms are of Latin origin. In current literature however, there is no consistent way of writing the term in italics or not. Because the leading journals in medicinal chemistry that are part of the American Chemical Society decided several years ago not to write these terms in italic letters any more, we now write it in normal letters in all our manuscripts (if the guidelines for authors don’t specifically state else).

The acronym for dopamine receptors (DA) can be DR or DAR, for me, DAR is more correct. But I assume that it is something debatable.

Again, the comment is absolutely correct and we agree that there is not only ONE correct way to abbreviate dopamine receptor. The most common versions include DR and DRD. Thus, we would prefer to keep the abbreviation DR.

Even if they are known, it is best to use the full description before using the acronym. For example, for FDA, SBP, ECFP4, TLB, NDF, and more.

We would like to thank the reviewer for the very detailed comment. As stated above, we now write the full name before using any abbreviation.

On the first two pages there are four paragraphs beginning with "However”. One suggestion is that the first "However" can be changed to "Moreover". Evaluate other possible changes.

The reviewer is absolutely right considering the over-use of however in the introduction. Thus, we´ve made some amends to the manuscript restructuring some of the sentences to improve linguistic quality.

What percentage of DMSO was used to dissolve the compounds?

All comments were initially dissolved in 100 % DMSO. This is now also stated appropriately in the materials and methods section. During in vitro assessments, a maximum of 1 % DMSO was allowed to ensure cell membrane integrity.

Some company names and/or addresses are missing. For example, who is VMR and what is his address? Tocris is a proper name, use a T; what is your address? The address of EVERPharma AT GmbH?

Tocris is indeed a proper name. Thus, we´ve corrected the spelling in the manuscript using a capital T. Considering the address of EVERPharma GmbH, the details were added to the manuscript. Considering the comment regarding VWR, the company changed its name meanwhile to Avantor, which is now used in the revised manuscript.

In section 2.9 use subscripts on dopamine receptors.

Again, the comment highlights the very thorough analysis of the manuscript conducted by the reviewer. We assumed that the comment refers to the names of the materials used during the in vitro assessments. In this particular case, subscripts were deliberately not used to reflect the appropriate naming of the materials in the way they are provided by the supplier. In any other case we agree with the reviewer that the respective receptor subtype has to be given using subscript.

For me, the most correct abbreviation is Ki (small i)

This is very true, thus the spelling was changed from KI to Ki in the text as well as in the figures and tables. The same changes were implemented considering changing KD to Kd to improve concistency.

Clozapine is not the representative of tricyclic antipsychotics. It is Chlorpromazine, compound 6.6.6, it was the first synthetic drug used as an antipsychotic / Clozapine is the prototype of the atypical antipsychotics, and all other atypical drugs are called clozapine-like.

This was a very valuable comment from reviewer 1. Accordingly, changes were implemented in the manuscript.

Reviewer 2 Report

This manuscript entitled “Dopamine Receptor Ligand Selectivity - An in silico/in vitro Insight” aims to address the selectivity issue of development of DR modulators, using in silico/in vitro combinatory workflow. This workflow is similar to authors’ previous paper, and indeed helps screen out some molecules showing selectivity toward subtype of DR. The workflow is solid and sound; therefore, this study is recommended published in this journal after minor revision.

1. The in silico/in vitro combinatory workflow is similar to authors’ previous paper, it is encouraging to step into further stage. All experiments in this study are in cell-based level, how about the outcome of investigated compounds in advanced level?

2. The sequence of section 2 is in disorder. The section of “Dataset assembly for molecular docking (ChEMBL validation)” should be 2.4. The following sections need modification, as well.

3. The format of reference is not consistent.

Author Response

We would like to thank reviewer 2 very much for the constructive feedback and the very thorough read-through of our manuscript. The comments helped us to further improve the quality of our manuscript. In the following we list the comments of reviewer 2 and answer to them accordingly.

The in silico/in vitro combinatory workflow is similar to authors’ previous paper, it is encouraging to step into further stage. All experiments in this study are in cell-based level, how about the outcome of investigated compounds in advanced level?

The reviewer is absolutely right considering this observation. The selectivity study in this manuscript was deliberately using a cell-based in vitro setup to be able to distinguish binding behaviour between the three different dopamine receptor subtypes investigated in this study. While it is true that it only shows a small piece of the puzzle it is the best way to get a detailed and specific readout considering subtype selectivity. To assess the true impact of the ligands´ selectivity, we do agree that a more physiological approach is needed to account for (poly-)pharmacological or off-target effects due to (simultaneous) binding to other molecular targets (in particular other aminergic GPCRs).

The sequence of section 2 is in disorder. The section of “Dataset assembly for molecular docking (ChEMBL validation)” should be 2.4. The following sections need modification, as well.

The sequence of section 2 was indeed disordered. We would like to thank the reviewer and note that we´ve corrected the order and numbering of the according section.

The format of reference is not consistent.

The reviewer is right. Despite using EndNote, the format was not consistent. The according amendments have been made.

Reviewer 3 Report

Dear authors,

I consider his work entitled "Dopamine Receptor Ligand Selectivity - An in silico / in vitro  Insight" is interesting and transcendent. However, major and minor changes are necessary to incorporate.

I share some comments about it that I hope will be useful:

** Major comments:

1.      Section 2.4: Not is clear if did you minimize the targets (PDB structures)? And, if did you remove the ligands containing it?

2.      Table 1 and 2: I understand that the selected amino acids to consider "flexible" are considered key interactions. But, What does not consider all binding sites as flexible?  // Exist a lot of work that demonstrates the relevance of flexibility amino acids that do not interact directly in the binding site on different targets. Maybe, some amino acids do not generate key interactions, but improve/facilitate the formation of key interactions. This "methodological issue" could generate binding scores and binding poses totally different to the presented results in this work.

3.      Section 2.4.2: Respectfully, I do not understand the methodological reasons to do not generating molecular dynamics calculations for all dopamine receptors studied by molecular docking.

4.      Section 2.4.2:  Please add a detailed description of each parameter and the protocol used to generate the molecular dynamics results. With the current description is impossible to reproduce their results. What is the water model? What are the barostat and thermostat used?  What type of ensemble did you use?, Etc.

5.      Section 2.4.2: How to validate their MD calculations? // Please, add the RMSD and RMSF plots of the control and ligands tested. What happened with the cavity studied? I.e,., did you observe changes in the binding site or changes in the general structure of the receptor? // For all these observations, respectfully, I strongly suggest redesigning the MD experiments or removing this section from the main text.

6.      Discussion section: Please talk about the possible off-target activity of these studied compounds against other neurological endpoints. For example, against targets like muscarinic or adenosine receptors, I share a reference that could be useful:  https://doi.org/10.12688/f1000research.124160.1

7.      All manuscript: Please add a conclusion section.

** Minor comments:

1.      Section 2.3: Is necessary explicitly mention the forcefield used to minimize all ligands.

2.      Section 2.6: Please, change "shown in Table 4", to "shown in Table 3".

Author Response

We would like to thank reviewer 3 very much for the constructive feedback and the very thorough read-through of our manuscript. We assume, that the reviewer has great expertise in the field of MD simulations due to the high quality of the comments. The comments helped us to further improve the quality of our manuscript. In the following we list the comments of reviewer 3 and answer to them accordingly.

Major comments:

Section 2.4: Not is clear if did you minimize the targets (PDB structures)? And, if did you remove the ligands containing it?

We thank the reviewer for the comment. In fact, the PDB structures were not minimized before or during the molecular docking process. However, originally bound ligands were extracted prior to docking. Amendments have been made in the manuscript stating the respective details.

Table 1 and 2: I understand that the selected amino acids to consider "flexible" are considered key interactions. But, What does not consider all binding sites as flexible?  // Exist a lot of work that demonstrates the relevance of flexibility amino acids that do not interact directly in the binding site on different targets. Maybe, some amino acids do not generate key interactions, but improve/facilitate the formation of key interactions. This "methodological issue" could generate binding scores and binding poses totally different to the presented results in this work.

First, we would like to thank the reviewer for this very important insight. We are aware of the impact of flexibility settings in different regions of the molecule and especially in the binding pocket itself. Setting the key interactions within the orthosteric binding pocket to flexible yielded very good results considering, first, the correlation between in silico and in vitro results and, second, allowed a precise discrimination od D2like-selective ligands. We absolutely agree, that an increased flexibility of selected amino acids, especially the ones contained within the secondary binding pocket, might increase the capability of the developed workflow to distinguish D2R- and D3R-selectivity. However, this approach would have been out of scope for this particular study where we deliberately wanted to focus on the in silico / in vitro correlation. More dynamic approaches will be focused on in follow-up projects to further investigate the discovered ligands and also develop in silico tools with enhanced predictive power.

Section 2.4.2: Respectfully, I do not understand the methodological reasons to do not generating molecular dynamics calculations for all dopamine receptors studied by molecular docking.

MD simulations prior to molecular docking can lead to improved performance, however, this is not always granted. Theoretical validation of a docking workflow is therefore used to determine if a native experimental structure is suitable for any given predictive calculation. Pre-processing all structures with MD prior to docking may unintentionally introduce another level of uncertainty regarding binding site geometries or protein-ligand interactions. Therefore, using the native crystallized structure out before further calculations may lead to valid and predictive workflows while also saving time in such a project. Initially, we tried docking into the cryo-EM structures for D1R, D2R and D3R, respectively. While for D1R and D3R, docking results were of high quality, the D2R docking proved to be more challenging. The position of docked apomorphine and other ligands would not correlate with information on the respective binding modes from scientific literature. Investigating the cryo-EM structures more closely, we found that the D1R and the D3R structures were bound to representative dopamine receptor ligands (apomorphine and PD-128907), while the D2R structure 7jvr was initially bound to bromocriptine. Since bromocriptine is a rather atypical ligand, we decided to simulate an induced fit binding site geometry of the D2R with apomorphine. First, we docked apomorphine into the bromocriptine-induced conformation of the binding pocket. We chose a binding complex, which corresponded to the expected protein-ligand interactions according to literature. Second, this apomorphine-including complex was the basis for the MD simulation. The thereby generated structure performed much better in our subsequent docking experiments and was therefore kept for further use.

Section 2.4.2: Please add a detailed description of each parameter and the protocol used to generate the molecular dynamics results. With the current description is impossible to reproduce their results. What is the water model? What are the barostat and thermostat used? What type of ensemble did you use?, Etc.

The input is highly appreciated. A summary of the minimization process settings is shown in Table 1. An overview of all parameters is given in Table 2. A summary of the Standard Dynamics Cascade as well as a detailed listing of all parameters is shown in Table 3 and Table 4. A Spherical Cutoff method is used for electrostatics. We used the standard parameters present in Biovia’s Discovery Studio.

Table 1. Summary of the Minimization process.

Minimizatio Criteria

CONJUG> Minimization exiting with number of steps limit (200) exceeded

Final RMS Gradient [kcal/(mol*A]

0.47676

Initial RMS Gradient [kcal/(mol*A]

1.16878

Electrostatic Energy [kcal/mol]

-74320.37289

Van der Waals Energy ]kcal/mol]

-7856.40360

Potential Energy ]kcal/mol]

-73078.46743

Initial Potential Energy [kcal/mol]

-70950.65436

Forcefield

CHARMm

Table 2. Detailed overview of the settings used during the Minimization process.

Minimization

Parameter

Setting

Algorithm

Smart Minimizer

Max Steps

200

RMS Gradient

0.1

Energy Change

00

Save Results Frequency

0

Implicit Solvent Model

None

Dielectric Constant

1

Implicit Solvent Dilectric Constant

80

Generalized Born Lambda Constant

-

Minimum Hydrogen Radius

0.8

Use Non-polar surface

True

Non-polar Surface Constant

0.92

Non-Polar Surface Coefficient

0.00542

Salt Concentration

0.0

Input Atomic Radii

van der Waals Radii

Use Molecular Surface

True

Nonbond List Radius

14.0

Nonbond Higher Cutoff Distance

12.0

Nonbond Lower Cutoff Distance

10.0

Electrostatics

Automatic

Kappa

0.34

Order

4

Minimization Constraints

-

Apply SHAKE Constraint

False

Number of Processors

1

Table 3. Summary of the Standard Dynamics Cascade process.

Final RMS Gradient [kcal/(mol *A]

0.650

0.075

19.257

19.232

19.127

Initial RMS Gradient [kcal/mol *A]

0.972

0.650

3.628

19.257

19.232

Electrostatic Energy [kcal/mol]

-74028.920

-75322.971

-74737.941

-80960.394

-81034.075

Van der Waals Energy [kcal/mol]

-7836.896

-7843.438

-6715.957

-6952.978

-6758.705

Temperature [K]

-

-

302.709

300.734

302.094

Kinetic Energy [kcal/mol]

-

-

13279.289

13192.649

13252.349

Potential Energy [kcal/mol]

-72964.947

-74257.521

-62155.902

-67927.536

-68117.446

Total Energy [kcal/mol]

-

-

-48876.612

-54734.887

-54865.097

Initial Potential Energy [kcal/mol]

-72896.919

-72964.947

-74257.521

-62155.902

-67927.536

End Time [ps]

-

-

4000

1004

1014

Start Time [ps]

-

-

0

4

1004

Forcefield

CHARMm

CHARMm

CHARMm

CHARMm

CHARMm

Stage

Minimization

Minimization 2

Heating

Equilibration

Production

Table 4. Detailed overview of the settings used during the Standard Dynamics Cascade process.

Standard Dynamics Cascade

Minimization

Parameter

Setting

Algorithm

Steepest Descent

Max Steps

1000

RMS Gradient

1.0

Constraints

-

Minimization 2

Parameter

Setting

Algorithm

Adopted Basis NR

Max Steps

2000

RMS Gradient

0.1

Constraints

-

Heating

Parameter

Setting

Simulation Time [ps]

4

Time Step [fs]

2

Initial Temperature

50.0

Target Temperature

300.0

Adjust Velocity Frequency

50

Save Results Interval [ps]

2

Constraints

-

Equilibration

Parameter

Setting

Simulation Time [ps]

1000

Time Step [fs]

2

Target Temperature

300.0

Adjust Velocity Frequency

50

Save Results Interval [ps]

2

Constraints

-

Production

Parameter

Setting

Simulation Time [ps]

10

Time Step [fs]

2

Target Temperature

300.0

Temperature Coupling Decay Time

5.0

Save Result Interval [ps]

2

Save Result File

True

Constraints

-

Type

NVT

TMass

1000.0

PMass

1000.0

PGamma

25.0

Reference Pressure

1.0

Implicit Solvent Model

Parameter

Setting

Implicit Solvent Model

None

Dielectric Constant

1

Implicit Solvent Dielectric Constant

80

Generalized Born Lambda Constant

-

Minimum Hydrogen Radius

0.8

Use Non-polar surface

True

Non-polar Surface Constant

0.92

Non-Polar Surface Coefficient

0.00542

Salt Concentration

0.0

Input Atomic Radii

Van der Waals Radii

Use Molecular Surface

True

Nonbond List Radius

14.0

Nonbond Higher Cutoff Distance

12.0

Nonbond Lower Cutoff Distance

10.0

Electrostatics

Automatic

Kappa

0.34

Order

4

Dynamics Integrator

Leapfrog Verlet

Apply SHAKE Constraint

True

Random Number Seed

314159 314159 314159 314159

Number of Processors

1

Section 2.4.2: How to validate their MD calculations? // Please, add the RMSD and RMSF plots of the control and ligands tested. What happened with the cavity studied? I.e,., did you observe changes in the binding site or changes in the general structure of the receptor? // For all these observations, respectfully, I strongly suggest redesigning the MD experiments or removing this section from the main text.

We would like to thank the reviewer for this very helpful comments regarding our MD simulation approach for the D2R. We also agree upon the comment, to remove the respective section from the main manuscript. However, we kept it in the supplementary information to keep the information as complete and transparent as possible.

Initially, all three DR structures were considered for MD simulations. Since apomorphine is used as a control in the discussed study, first, we docked apomorphine into all of the chosen cryo-EM structures of D1R, D2R and D3R. Since the D1R structure (PDB entry 7jvq) is originally bound to apomorphine and the D3R structure (PDB entry 7cmv) is bound to PD-128907 (a similar structure to apomorphine) the docking results were satisfying (shown in Figure 1a and b) without further modifying the protein structures using MD.

Figure 1.

In contrast, from a structural point of view, bromocriptine is a very different ligand (also compared to apomorphine and PD-128907) and induces a completely different geometry of the receptor binding pocket. This is highlighted in Figure 2, where we docked apomorphine into the unaltered D2R cryo-EM structure (PDB entry 7jvr) originally bound to bromocriptine.

Figure 2. Docking of apomorphine in the pre-MD structure of D2R.

In comparison to the docking results shown in Figure 1 the posing is more heterogeneous where only a single docking pose is oriented correctly (highlighted in red) within the binding pocket. Thus, the D2R structure was considered the perfect candidate for a MD simulation. Comparing the binding pockets of the original cryo-EM structure (shown in red in Figure 3a and b) and the MD-modified structure bound to apomorphine (shown in green in Figure 3b) the impact of bromocritpine on the geometry of the binding pocket is highlighted even better.

Figure 3. Comparison of the binding pockets (red, PDB entry 7jvr; green, MD-modified structure) of D2R bound to (a) bromocriptine and (b) apomorphine after the MD simulation.

While the cavity is extended in the bromocriptine-bound state it shrinks drastically after docking apomorphine and performing the MD simulation. The success of the MD simulation is also shown by the increase of correct docking poses (29 out of 30 poses) and increased pose homogeneity shown in
Figure 4.

Figure 4. Docking of apomorphine in the post-MD structure of D2R.

Discussion section: Please talk about the possible off-target activity of these studied compounds against other neurological endpoints. For example, against targets like muscarinic or adenosine receptors, I share a reference that could be useful:  https://doi.org/10.12688/f1000research.124160.1

We thank the reviewer very much for the excellent reference which we´ve included and cited in the discussion section of the manuscript. The reviewer is completely right, that polypharmacological as well as possible off-target effects are extremely important when investigating the discussed ligands. However, to generate reliable in silico and also in silico / in vitro correlation data it is important to initially isolate the ligand-receptor interactions. We think that only after understanding the separate selectivity mechanisms it is possible in add-on studies to increase the complexity investigate multiple ligand-receptor interactions at once (especially in an in silico setting).

All manuscript: Please add a conclusion section.

A conclusion section was added to the manuscript highlighting the most interesting findings of the study.

Minor comments:

Section 2.3: Is necessary explicitly mention the forcefield used to minimize all ligands.

The minimization performed in OMEGA used the mmFF94 forcefield. Changes have also been implemented into the manuscript.

Section 2.6: Please, change "shown in Table 4", to "shown in Table 3".

The respective references have been changed in the manuscript.

Round 2

Reviewer 3 Report

Dear Authors,

I consider that your work "Dopamine Receptor Ligand Selectivity - An in silico / in vitro Insight" has interesting and transcendent. In my opinion, your manuscript is ready to publish in its present form.

Best regards.